# Case Study of Chosen Sandwich-Structured Composite Materials for Means of Transport

**Pavel Koštial [1], Zora Koštialová Jančíková [1], Ivan Ružiak [2,*] and Milada Gajtanska [2]**

[1]  Faculty of Materials Science and Technology, VŠB–Technical University of Ostrava, 17. listopadu 15/2172, 708 33 Ostrava-Poruba, Czech Republic; pavel.kostial@vsb.cz (P.K.); zora.jancikova@vsb.cz (Z.K.J.)

[2]  Faculty of Wood Sciences and Technology, Technical University in Zvolen, T. G. Masaryka 24, 960 01 Zvolen, Slovak Republic; gajtanska@tuzvo.sk

\*  Correspondence: ruziak@tuzvo.sk; Tel.: +421-45-5206470

**Abstract:** Modern means of transport increasingly utilize sandwich constructions. Among other things, the reasons for such state of affairs include the reduced weight of means of transport, and through this, better fuel economy as well as price. This work is dedicated to a systematic experimental study of the influence of various materials and sandwich designs on their mechanical properties. In the framework of experiments, sandwich-structured composites were exposed to two types of stressing: static as well as impact stressing. The testing of prepared samples was performed according to ASTM C-393 Standard, dealing specifically with the bending behavior of sandwich composite constructions and impact testing under the scope of ISO 6603-2 Standard test. In this article we deal with static and impact testing of the eight types of core materials, two types of coatings, two types of surface finishes, and two types of resins with a special emphasis on their use in constructions of some exterior or interior components of transport means.

**Keywords:** sandwiches; mechanical properties; polymers; honeycomb; prepregs; transport means

## 1. Introduction

Because of their specific properties, sandwich constructions are widely used in the industry of transport means, particularly because of the reduced weight of such products. It stands to reason that their first applications appeared in the aircraft industry. Favorable properties accelerated their further development, and now there are unlimited quantities of such materials available on the market. The incessant development of new materials leads to gradual reductions in their prices, which makes sandwich constructions affordable in other transport applications (mainly in ship-building and railway branches), mainly because of another advantage—specific rigidity. This is rather advantageously exploited in large-area products, i.e., for various cladding purposes (either interior or exterior, ceiling panels, bulkheads, floor panels, etc.). If composite shaping properties for coating of sandwich constructions are used, then the sandwich material may represent even the product design. Typical representatives of purely design applications are front sections of railway units, where shaped polymer foams are used as a core in combinations with composite polyester–glass coatings.

However, one of the main disadvantages of these constructions is their price. This is caused not only by the high prices of input materials, but also by demanding manufacturing processes.

The influence of fiber length and fiber orientation on damping and stiffness of polymer composite materials is in details described in [1]. The utilization of special aluminium foams in sandwich structures is described in [2]. The impact behavior of honeycomb structures is described in [3]. Ecofriendly materials for sandwich structures were studied from an acoustical point of view in [4,5].

In [6–13], the authors present further sandwich structures. The results of diagnostic methods suitable for sandwiches and their components are described in [14–24].

Of course not only weight and rigidity are the major prerogatives of sandwich constructions. Their further advantages consist not only in combinations with other materials, thermal insulation, and noise attenuation properties, but also in their fire resistance, which plays a significant role in the whole transport branch.

The work of Kamiński [25] solves the problem of multiscale homogenization of n-component composites with semi-elliptical random interface defects.

A series of self-matting waterborne polyurethanes were successfully prepared by introducing hydrophilic units into both soft and hard segments. By employing a polycaprolactone polyol containing carboxylate groups within the polymer chains to provide hydrophilicity directly, the matting performance of the studied films was greatly improved in [26].

The authors of [27] consider field theory formulation for directed polymers and interfaces in the presence of quenched disorder.

The fabrication of a sandwich-structured specimen with different material combinations using conventional thermoplastics such as polylactic acid, acrylonitrile butadiene styrene, and high impact polystyrene through the filament-based extrusion process can demonstrate an improvement in its properties [28]. Among these materials, this paper aims to assess the best material sandwich-structured arrangement design, to enhance the mechanical properties of a part and to compare the results with the homogeneous materials selected.

Sound absorbing composites with stratified structures, including double-layer and sandwich structures, were studied through the combination of nitrile butadiene rubber and polyurethane foam. The thermal properties of the materials studied in this paper can be found in [29]. Other authors have studied the thermal and acoustical properties of wood wool-based materials [30], which together with PU foam/NBR rubber composites are also used as thermal and acoustical insulations.

An artificial neural network was used to investigate the influence of the core density and number of carbon fiber-reinforced polymer layers on the mechanical properties. The results showed an improvement of specific strength and elastic modulus with increasing number of carbon fiber-reinforced polymers [31]. The thermal properties of the materials studied in this paper can be found in [32].

In this work, a wide range of experiments concerning the selected mechanical properties of specific sandwich materials with a special emphasis on their use in constructions of some exterior or interior components of transport means is presented. In order to meet this goal, material tailoring was used as an alternative way in the sandwich design. Eight types of core materials, two types of coatings, two types of surface finishes, and two types of resins were experimentally tested.

## 2. Materials and Methods

As the basic material for coating, glass-phenolic prepreg, PH840-300-42 type, was selected. It is used mostly in aviation and railroad industries for interior applications.

For coat hybrid types, epoxy-aramid, IMP 530 type was used. Typical properties of the used these prepregs are shown in Table 1.

**Table 1.** Coating material characteristics.

| Marking | Reinforcement Type | Matrix Type | Matrix Content (%) | $T_g$ (°C) | Bending Strength (MPa) | Bending Modulus (GPa) |
|---|---|---|---|---|---|---|
| PH840-300-42 | glass/fabric sateen/280 g·m$^{-2}$ | phenolic | 42 | >80 | 450 | 22 |
| IMP 530 | aramid/fabric linen/200 g·m$^{-2}$ | epoxy | 42 | 120 | 148/170 | 35–40 |

An aramid paper honeycomb, Nomex® type T 722 (Schűtz, Selters, Germany), was selected as the core material. Nomex paper honeycombs have low specific densities, and high thermal stability and fire resistance [24].

Another material used as the core material was PET (polyethylene terephthalate) foam supplied under the AIREX T90-100 trademark (AlcanAirexcomposites, Sins, Switzerland). This is a polyethylene terephthalate closed-cell foam featuring a good fatigue life, very good thermal stability and resistance against chemicals, and the foam is also recyclable.

The last type of core material is an ECM-type aluminium honeycomb (Eurocomposites, Echternach, Luxembourg). The specific properties of the individual core materials are shown in Table 2.

As regards the honeycomb hexagonal structure (Figure 1), its mechanical properties are not equal in all the directions. Therefore, the following markings of honeycomb (HC) dimensions were introduced.

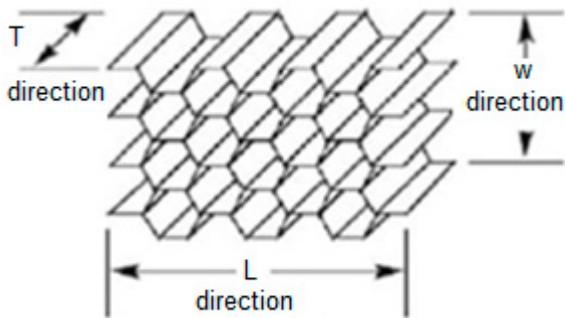

**Figure 1.** Honeycomb design.

**Table 2.** Material characteristics of core materials.

| Marking | Material Type | Cell Size (mm) | Density (kg·m⁻³) | Shear Strength L/W-Direction (MPa) | Shear Modulus L/W-Direction (MPa) |
|---|---|---|---|---|---|
| C2-3,2-32 | Nomex® T 722 | 3.2 | 32 | 0.6/0.42 | 17/25 |
| C2-3,2-48 | Nomex® T 722 | 3.2 | 48 | 1.2/0.7 | 36/24 |
| C2-3,2-64 | Nomex® T 722 | 3.2 | 64 | 1.6/1.95 | 50/35 |
| C2-3,2-80 | Nomex® T 722 | 3.2 | 80 | 1.75/1.0 | 56/32 |
| C2-4,8-64 | Nomex® T 722 | 4.8 | 80 | 1.6/0.85 | 50/27 |
| ECM-4,8-77 | Al alloy 5003 (AlMnCu) | 4.8 | 77 | 2.25/1.52 | 456/265 |
| AIREX T90-100 | PET | - | 100 | 0.8 | 20 |

Surfaces of sandwich constructions were covered by two types of paint systems used for interior applications in the rolling stock branch. Low-pressure wet painting was use as the coating process. The following surface finishes were used.

- Surface finish No. 1: Filler: Nuvovern Primer: acrylic–polyurethane-based anticorrosion coating with phosphate pigments (Mäder Läcke, Killwangen, Switzerland) Topcoat: Nuvovern ACR Emaillack-two-component polyurethane coating (Mäder Läcke, Killwangen, Switzerland)
- Surface finish No. 2: Filler: H/S FillingPrimer 463-5A (Mankiewicz, Hamburg, Germany) Topcoat: Texture Paint 476-21 (Mankiewicz, Hamburg, Germany)

In the diagrams and illustrations, the surface finishes are marked as SF1, SF2, or, as the case may be, without SF. The types of tested sample constructions are shown in Figure 2.

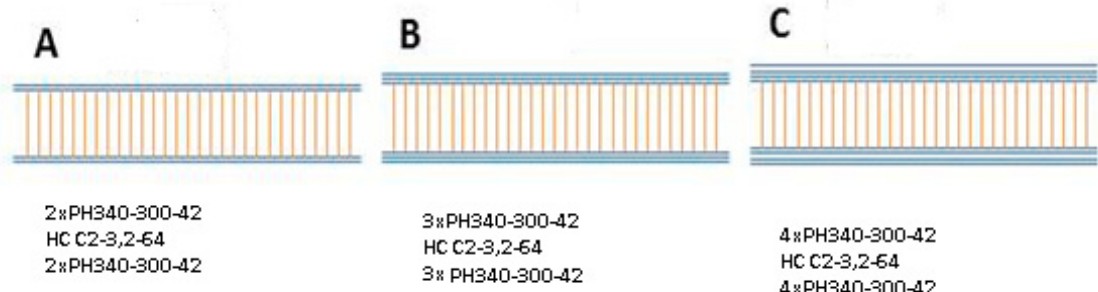

**Figure 2.** Constructions of basic samples. **A**: Honeycomb C2-3.2-64 is coated with two layers od PH340-300-42 coatings on both sides; **B**: Honeycomb C2-3.2-64 is coated with three layers od PH340-300-42 coatings on both sides; **C**: Honeycomb C2-3.2-64 is coated with four layers od PH340-300-42 coatings on both sides.

Methods

Eight types of core materials, two types of coatings, two types of surface finishes, and two types of resins were experimentally tested by 3pt bending test according to ASTM C-393 standard and impact tested according to ISO 6603-2 Standard test. The tests were provided by five samples of the same construction and of the same materials and bending strength $\sigma_{fM}$, bending modulus **E** were computed from the stress-strain diagram. From 5 values of this mechanical properties we have computed the mean value and standard deviation (which are plotted in Figure 4, Figure 5, Figure 6, Figure 7, Figure 8, Figure 9, Figure 10, Figure 11, Figure 12, Figure 13, Figure 14, Figure 15, Figure 17 and Figure 18) using the ZWICK 1456 test equipment (ZwickRoell, Brno, Czech Republic). In the second step the materials were impact tested according to ISO 6603-2 norm and maximal force $F_{max}$ obtained at 30 J and 60 J impact energy were measured. The 5 values for each material were subsequently processed by descriptive statistics parameters namely mean value and standard deviation which are also plotted in the Figure 19, Figure 20 and Figure 21. From these two statistical parameters, we have computed the coefficient of variation, which for all measuring samples and all measured properties was lower than 15%, which is an acceptable level of error. The figures show statistically significant differences; therefore, ANOVA tests results are not shown in this study. The shape of the samples was adapted to the requirements of the relevant standard.

Paper objectives

The objectives of the paper was to find whether the sandwich-structured coated composites can be used as parts in constructions of some exterior or interior components of transport means. Therefore chosen sandwich-structured coated composites were mechanically tested and bending strength, specific bending strength (ratio of bending strength to density), bending modulus and specific bending modulus (ratio of bending modulus to density) at static testing under ASTM C-393 framework and impact testing at 30 J and 60 J impact energy with computed mean values of maximal obtained force. Computation of specific bending strength and specific bending modulus is crucial for the application of materials in the means of transport because of lowering price and better fuel economy.

A comparison of mass balance of sandwich constructions to alternative materials of various thicknesses is shown in Figure 3.

While the sandwich panels were produced by the Vacuum Bag Technology, curing of sandwich constructions was carried out in the LAC, type SV11900/25 cure oven, using the basic curing mode of 3 h at 130 °C. In order to provide the ideal impregnation conditions for coating and the creation of bonds between the honeycomb and the coat (i.e., the capillary lift of resin along the cell walls), the whole rise-up time to this temperature took 60 min. A detailed flow chart of the curing process is shown in Figure 4. In order to prevent excessive resin losses in the construction during the impregnation process, the Wrighlon 3900 separation film (30 μm thick, total perforation per area of 0.14%, Airtech International,

Huntington Beach, CA, USA) was used. Once the panel was produced in this way, individual test samples were prepared from it by water jet cutting to appropriate sizes.

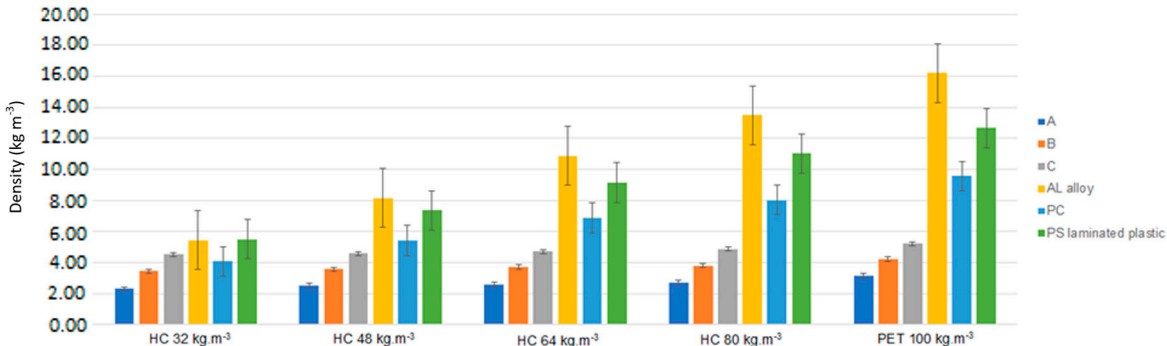

**Figure 3.** Mass balance of tested constructions. A comparison with other materials used in the sphere of production of transport means.

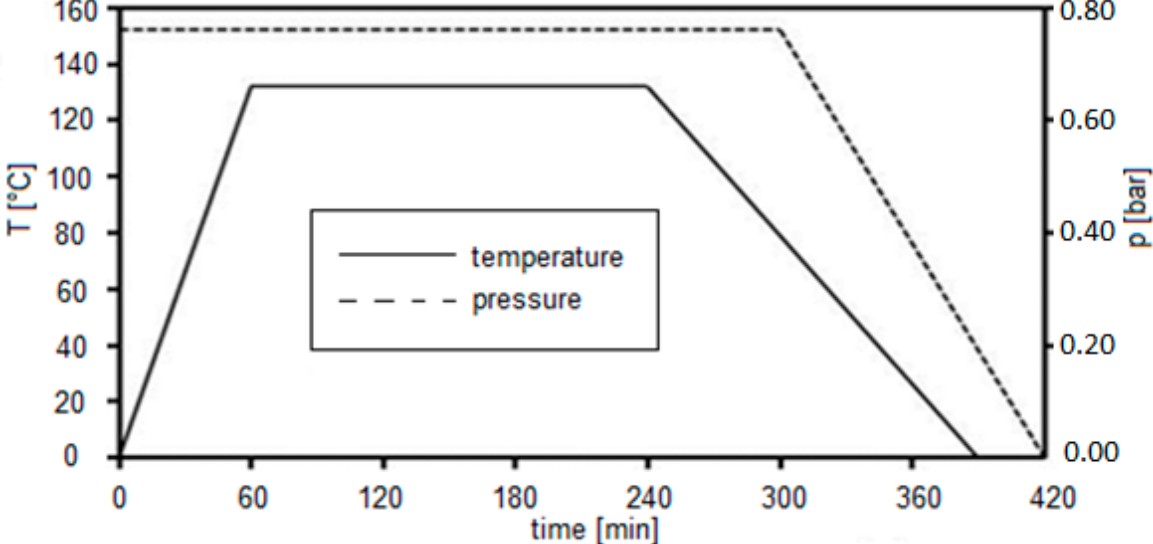

**Figure 4.** Curing cycle 1 in the sample production.

Sandwich-structured composites were tested in the experimental framework by the following stress methods.

- Static stressing (three-point bending).
- Impact stressing (dynamic impact tests).

## 3. Results

### 3.1. Static Testing—Core Type Influence

One of the experiment's main objectives was to monitor the influence of the coat thickness or, as the case may be, number of laminated layers, on their basic mechanical properties. The bending strength of the A-construction basic beam consisting of the honeycomb core "coated" at each side by two layers of selected prepregs is taken into consideration. By increasing the number of layers in the construction, both the mechanical and physical properties were changed over an extensive range. In the research first phase, a honeycomb produced from Nomex paper with the core thickness of 8 mm and unified geometric structure (3.2 mm mesh size) was selected for observing the influence of coating thickness on the bending characteristics.

PET foam AIREX T90 of the same thickness of 8 mm and density of 100 kg·m$^{-3}$ was used as another potential core material for sandwich constructions. The influence of coating thickness was investigated on honeycombs of various densities (32, 48, 64, and 80) kg·m$^{-3}$.

The course of expected growth in the strength by increasing of coating thickness is shown in Figure 5. Bending strength differences between the A and B constructions fluctuate from 21.95% to 38.77%, while another growth in the strength for the C-type constructions was lower, from 2.85% to 16.27%.

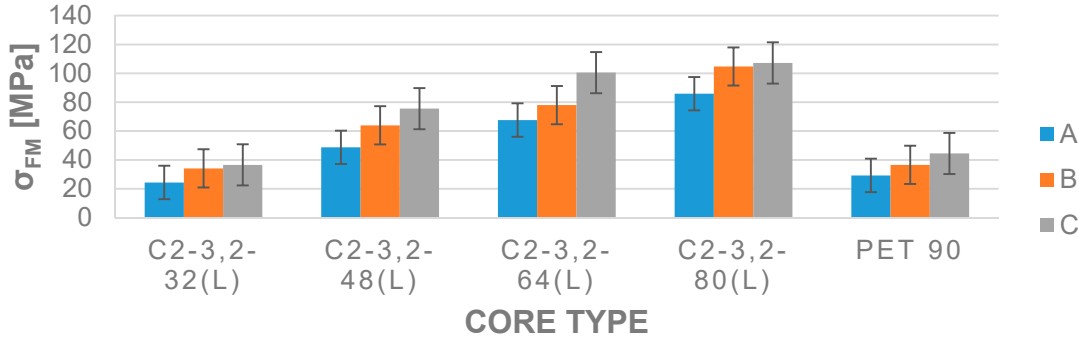

**Figure 5.** Influence of A–C coatings on the strength of the sandwich constructions. Core used: C2 honeycomb with unified mesh size of 3.2 mm and various densities (32, 48, 64, 80) kg·m$^{-3}$, orientation: L and PET foam (80 kg·m$^{-3}$).

If converted to the total density of sandwich constructions, it is evident that the strength growth by increasing the coating thickness is not efficient in terms of weight (with increasing the core thickness, the specific strength decreases, particularly in the case of C-type constructions). A survey of specific strengths in the observed constructions is shown in Figure 6.

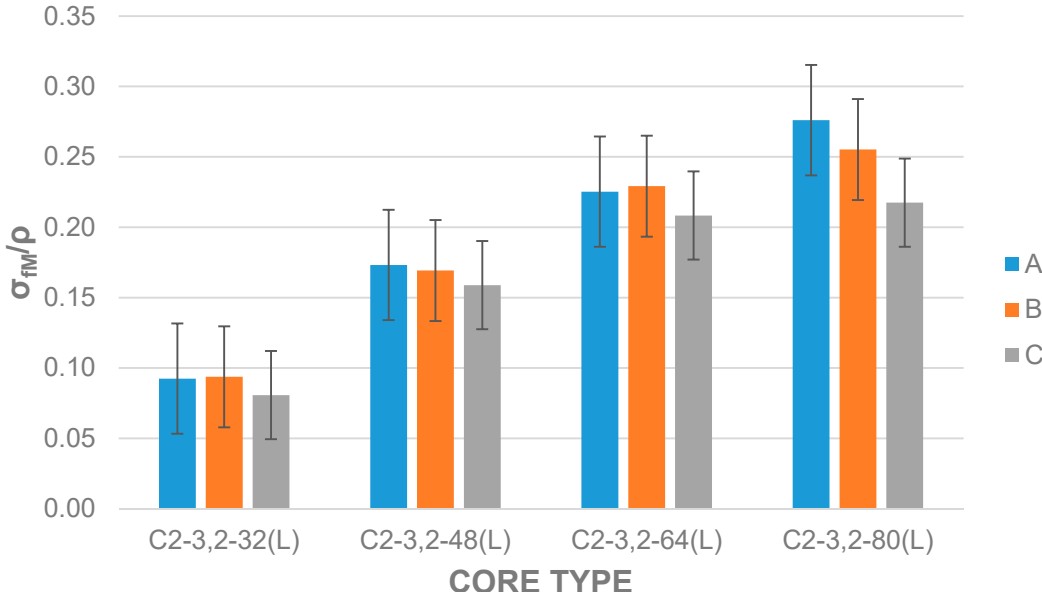

**Figure 6.** Specific strengths of sandwich constructions (A–C). C2-type honeycomb used with 3.2 mm individual mesh size and various densities (29, 48, 64, and 80) kg·m$^{-3}$, honeycomb orientation: L.

As the coating thickness increases, the bending modulus value ($E_O$) increases as well (Figure 7); in a comparison with the A-construction, the B-construction shows an average increase by 26.81%, while the C-constructions (as compared to the B-construction) show a lower growth at the level of 11.57%.

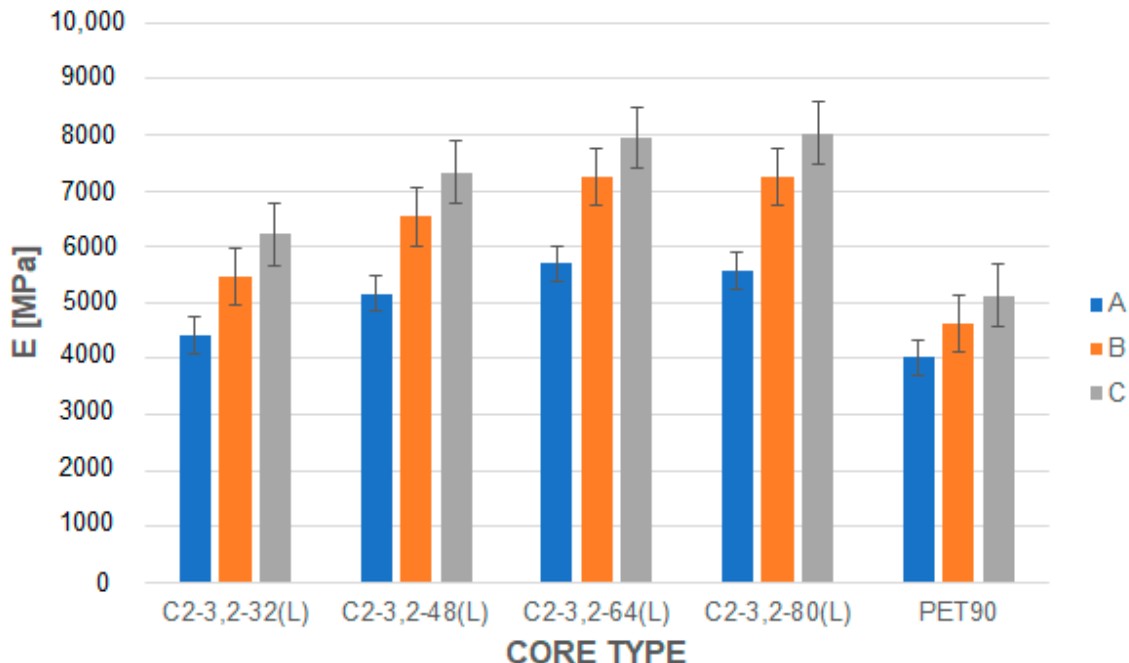

**Figure 7.** Influence of coating types (A–C) on the effective bending modulus of sandwich constructions; core used: C2 type honeycomb with unified mesh size 3.2 mm and different densities (32, 48, 64, and 80) kg·m$^{-3}$, L orientation, and PET foam (90 kg·m$^{-3}$).

Specific bending modulus E/$\rho$ decreases as the coating thickness increases (decrease t was noted in C-type constructions was more significant) and, in general, it particularly depends on the core materials used (Figure 8).

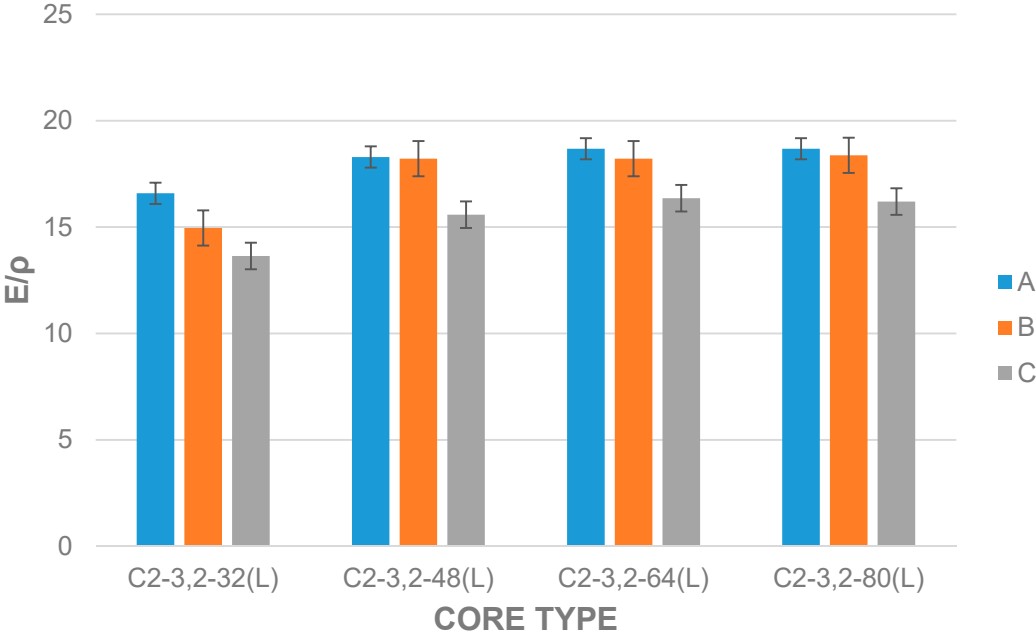

**Figure 8.** Specific bending modulus of sandwich constructions (A–C); honeycomb used: C2 type, C2 type honeycomb with unified mesh size 3.2 mm and different densities (32, 48, 64, and 80) kg·m$^{-3}$, L orientation.

As regards sandwich construction strength characteristics in terms of the honeycomb density, there is a significant positive effect on the bending strength (see Figure 9). As the honeycomb density

changed by 16 kg·m$^{-3}$, its strength increased in the A-type constructions by 55.18%, while in the B and C constructions, strength increased, on average, by 48.69 and 48.83% respectively, with the greatest difference in strength occurs between honeycomb densities 32 kg·m$^{-3}$ and 48 kg·m$^{-3}$; however, with the increase in honeycomb densities, such differences decrease.

On the other hand, with the use of honeycombs of higher densities, the bending modulus has minor effects: with the increase in honeycomb density, an average growth in the bending modulus in A-type construction by 8.24% was noted, while the same values for B- and C-type constructions were 10.07% and 9.04%, respectively. Changes in bending moduli with regard to changing honeycomb densities are shown in Figure 10.

By using higher density honeycombs, it is possible to significantly increase the strength of sandwich constructions with negligible increases in their weight.

Apart from the influence on the strength, the honeycomb density also strongly influences its workability; therefore, the honeycomb density is a significant parameter in the designing and manufacturing processes of sandwich constructions.

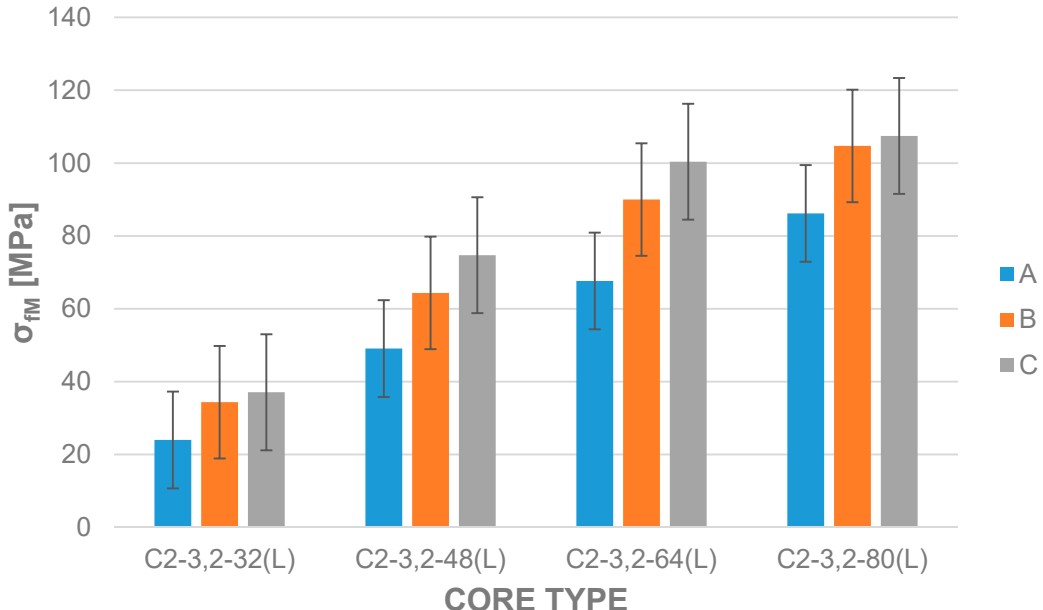

**Figure 9.** Influence of honeycomb density (32, 48, 64, 80) kg·m$^{-3}$ on the bending strength of sandwich constructions A–C (L orientation).

In evaluations of the influence of honeycomb cell sizes, constructions made with honeycomb of 64 kg·m$^{-3}$ density and cell sizes of 3.2, 4.8, and 6.4 mm were compared. Regarding the evaluated A, B, and C constructions, the highest values of strength were found at the constructions with the cell size of 4.8 mm, while in the constructions with the cell size of 6.4 mm, the values were lowest. A similar trend was noted for the bending modulus.

In terms of the coating influence effectivity in the sandwich constructions in the L direction, the average increases in strength of 22.81% (3.2 mm mesh size), 25.32% (4.8 mm mesh size), and 21.44% (6.4 mm mesh size) were recorded; in the W direction, the strength increased only slightly by 6.21% (3.2 mm mesh size), 16.61% (4.8 mm mesh size), and 6.26% (6.4 mm mesh size).

The development of strength and bending modulus in terms of honeycomb geometry are shown in Figures 11 and 12.

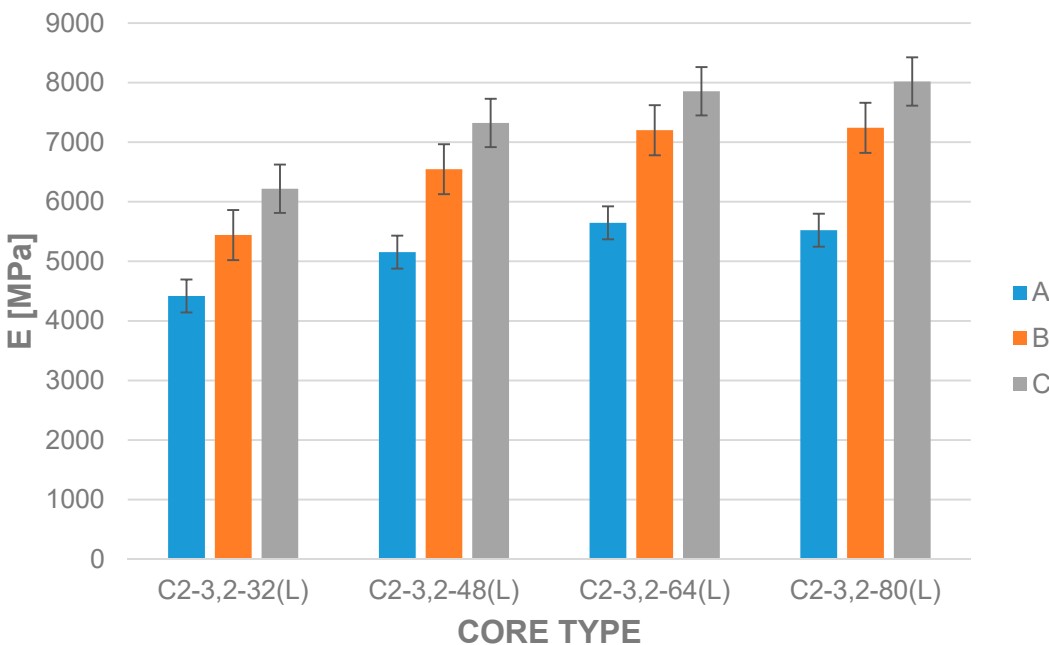

**Figure 10.** Influence of honeycomb density (32, 48, 64, 80) kg·m$^{-3}$ on the bending modulus of sandwich constructions A–C (L orientation).

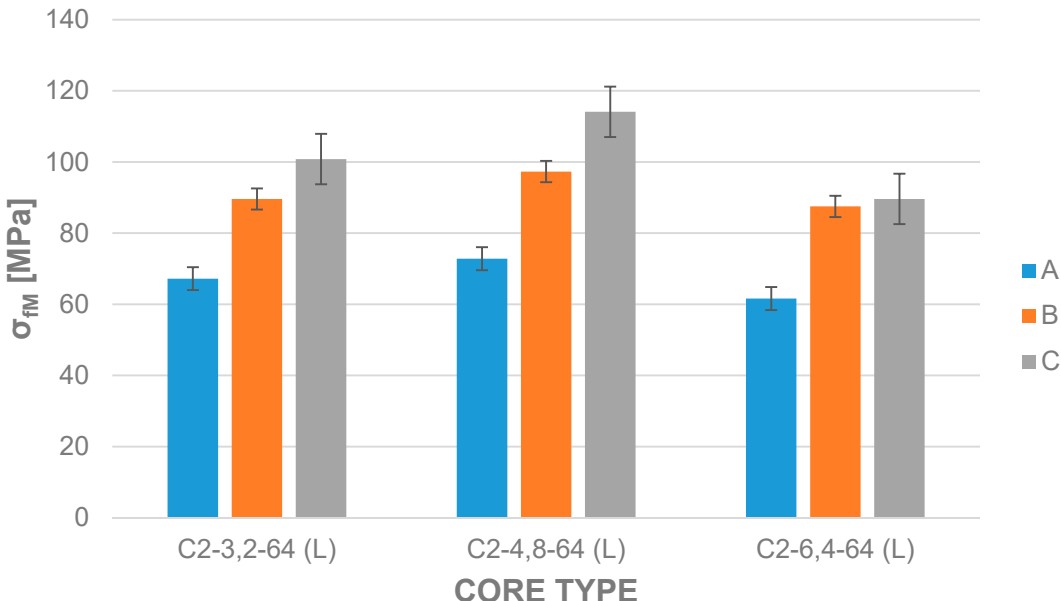

**Figure 11.** Influence of honeycomb geometry with mesh size (3.2, 4.8, and 6.4) mm on the bending strength of sandwich constructions A–C.

An overall survey of bending strength and its comparisons in both directions is shown in Figure 13. Differences between strengths in the W and L directions fluctuated with honeycomb densities (32, 48, and 80) kg·m$^{-3}$ in relatively equal values: 74.07% (32 kg·m$^{-3}$ density), 74.38% (48 kg·m$^{-3}$ density), and 76.39% for 80 kg·m$^{-3}$ density. Only the honeycombs with the density of 64 kg·m$^{-3}$ showed the difference at a lower level: 31.31%. The highest difference between the bending strengths in the W and L directions was found at the honeycomb density of 64 kg·m$^{-3}$.

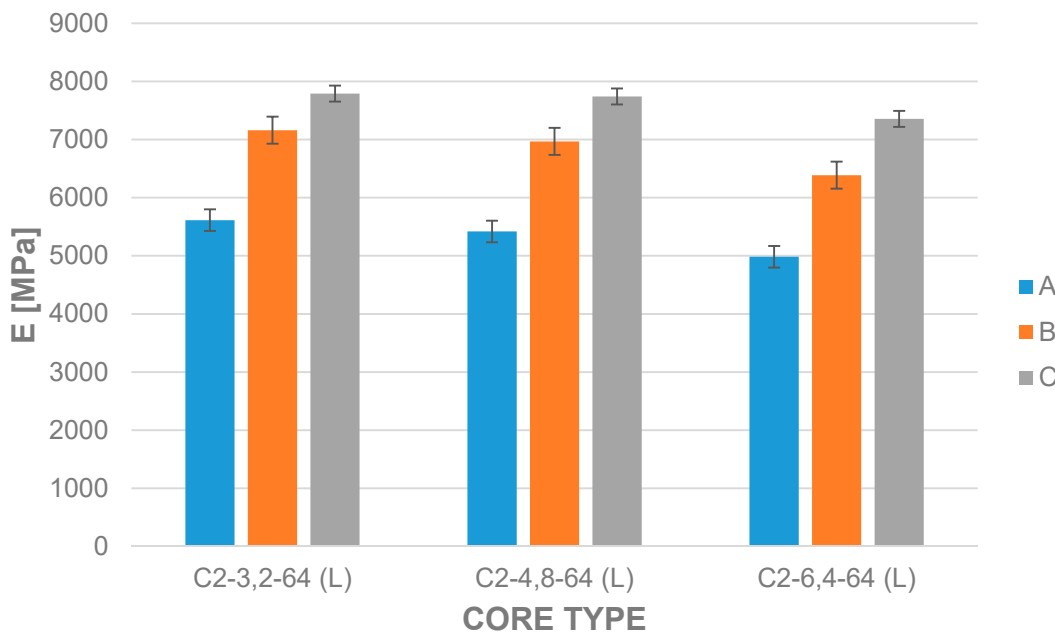

**Figure 12.** Influence of honeycomb geometry with mesh size (3.2, 4.8, and 6.4) mm on the bending modulus of sandwich constructions A–C.

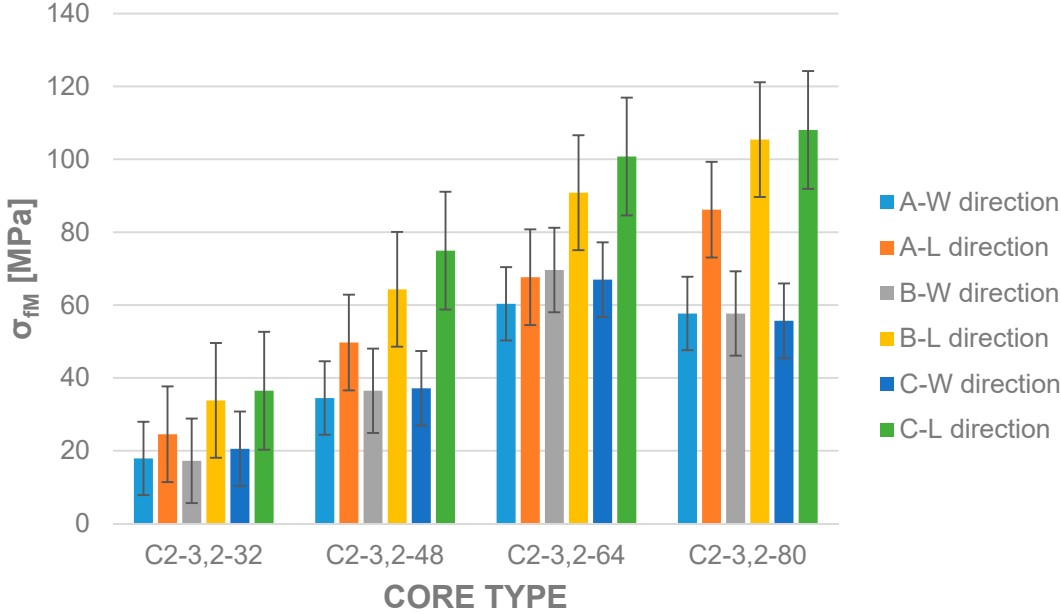

**Figure 13.** Influence of honeycomb orientation (L vs. W direction) on the bending strength of sandwich constructions (A–C types—overview).

The honeycomb orientation in sandwich constructions affects the bending modulus in a like manner. The effective bending modulus in all constructions is larger in the L direction than in the W direction, but the differences are more variable: for the density of 29 kg·m$^{-3}$, the average difference is 53.57%; for 48 kg·m$^{-3}$, the average difference is 26.05%; for 64 kg·m$^{-3}$, the average difference is the smallest (8.86%); and for 80 kg·m$^{-3}$, a difference of 17.26% was observed. The undeniably smallest difference was observed in the B-type constructions using the honeycomb density of 64 kg·m$^{-3}$, while the highest difference was found with the B-type constructions combined with the honeycomb density of 32 kg·m$^{-3}$, reaching the level of 58.56%. Therefore, it can be stated that the differences in the bending moduli between the L and W directions are decreasing with increasing honeycomb densities.

A survey of the bending moduli in W directions and their comparison with L directions is shown in Figure 14.

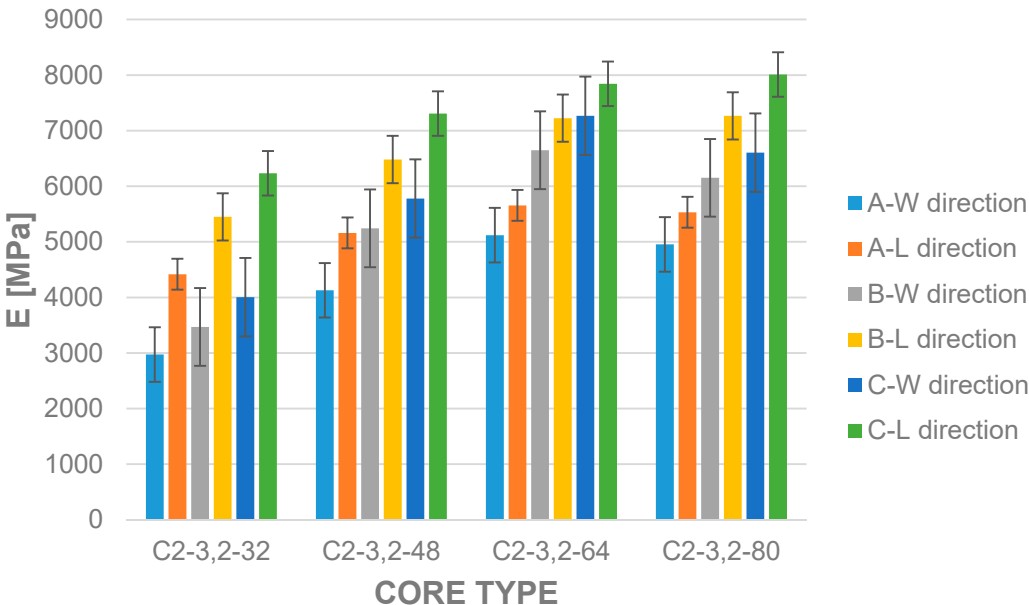

**Figure 14.** Influence of honeycomb orientation (L and W directions) of various densities on the effective bending modulus of sandwich constructions: types A–C (overview).

The influence of honeycomb orientation on the bending characteristics was evaluated in terms of another parameter of honeycomb geometry, i.e., the mesh size. In this case, a constant density honeycomb (64 kg·m$^{-3}$, with mesh sizes of 3.2, 4.8 and 6.4 mm) was evaluated.

Likewise, as in the previous case, the bending strength in the L direction was always higher than the bending strength in the W direction (Figure 15). As the strength in the W direction increases only slightly with the coating thickness, the differences in the bending strength between L and W increase with increasing coating thickness.

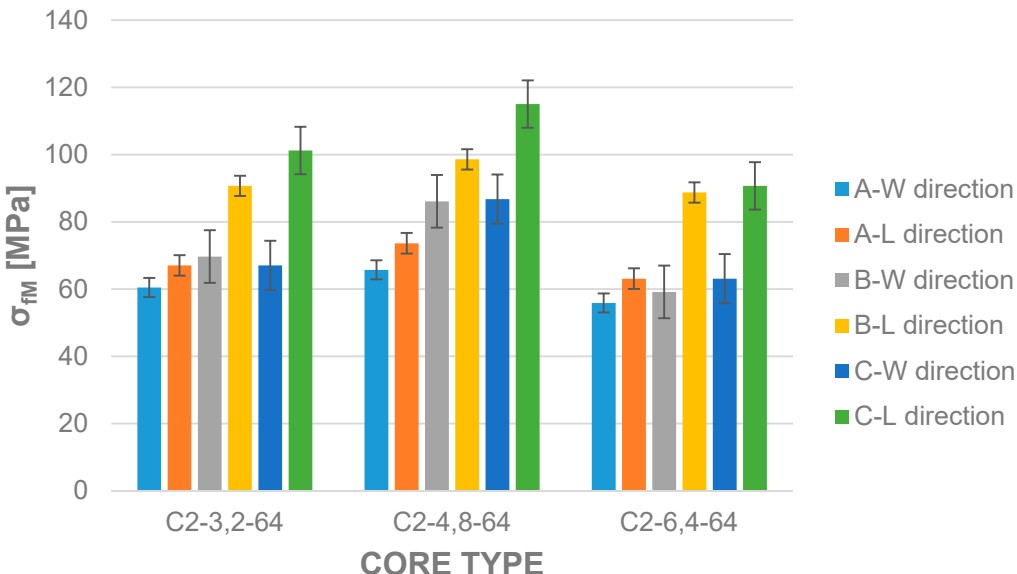

**Figure 15.** Influence of honeycomb orientations (L and W) with various geometry (mesh sizes 3.2 mm, 4.8 mm and 6.4 mm) on the bending strength of sandwich constructions (types A–C).

Regarding the bending modulus, the situation remains the same. The bending modulus magnitude in the W direction, for all three observed cell sizes (3.2, 4.8 and 6.4 mm), copies the bending modulus values in the L direction, but of course at lower value levels. The relevant survey is shown in Figure 16.

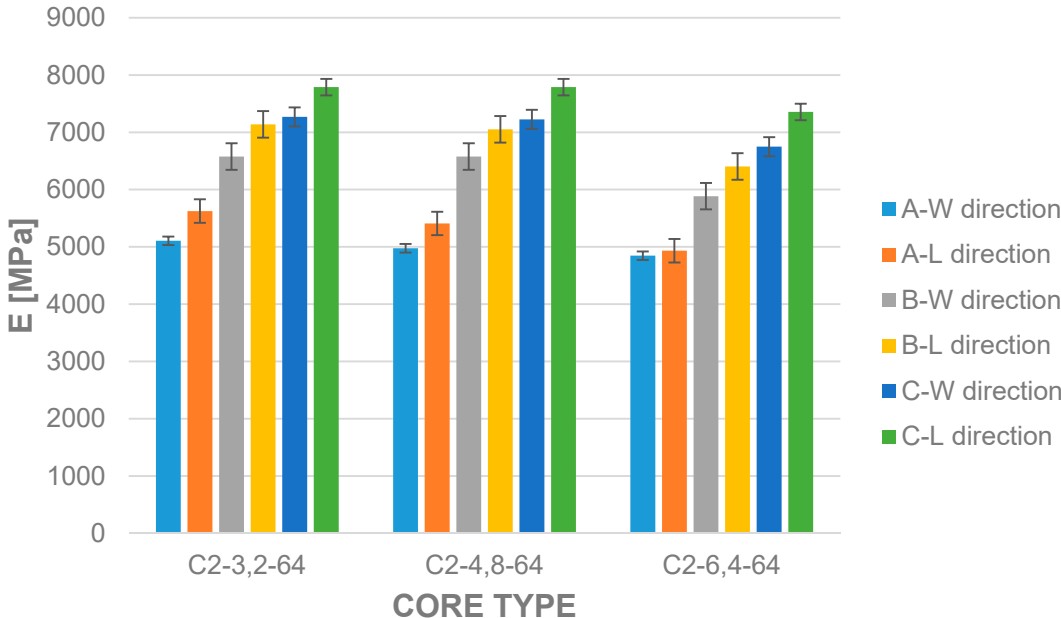

**Figure 16.** Influence of honeycomb orientations (L and W) with various geometry (mesh sizes 3.2 mm, 4.8 mm, and 6.4 mm) on the bending modulus of sandwich constructions (types A–C).

### 3.2. Static Testing—Construction Type Influence

Temperature modes and curing conditions for composite materials represent important process parameters that affect their final mechanical properties. Because the studied sandwich constructions are produced in a single process step with the composite coating, the curing conditions of the composite polymer matrix may consequently influence the properties of the whole sandwich construction. In our sandwich constructions, no additional film was used and therefore the very matrix of the composite coating (prepreg) had a substantial role in the creation of bonding between the coating and the core.

As the temperature increases, the viscosity of the phenolic resin drops, and between 80 and 90 °C, the resin starts to flow and creeps up along the surfaces of the honeycomb cell walls. It is therefore advisable to adjust the temperature mode in a way that allows the transport of a part of the polymer resin from the prepreg at certain time to the interface of coating and glass fabric/honeycomb. In the production of composites for sandwich constructions, not only is the primary goal for the best reinforcement matrix ratio in the composite important, but also the same importance must be put on strength optimization at the coating–core interface. For this reason, the influence on bending characteristics in the monitored sandwich constructions was observed in three curing cycles. Their developments are illustrated in Figures 17 and 18. The influence of curing temperatures on the strength of each monitored construction differed from case to case. In the A and B constructions the highest strength was recorded with the Cycle 1 (130 °C); however, the C construction showed the lowest level in this very cycle.

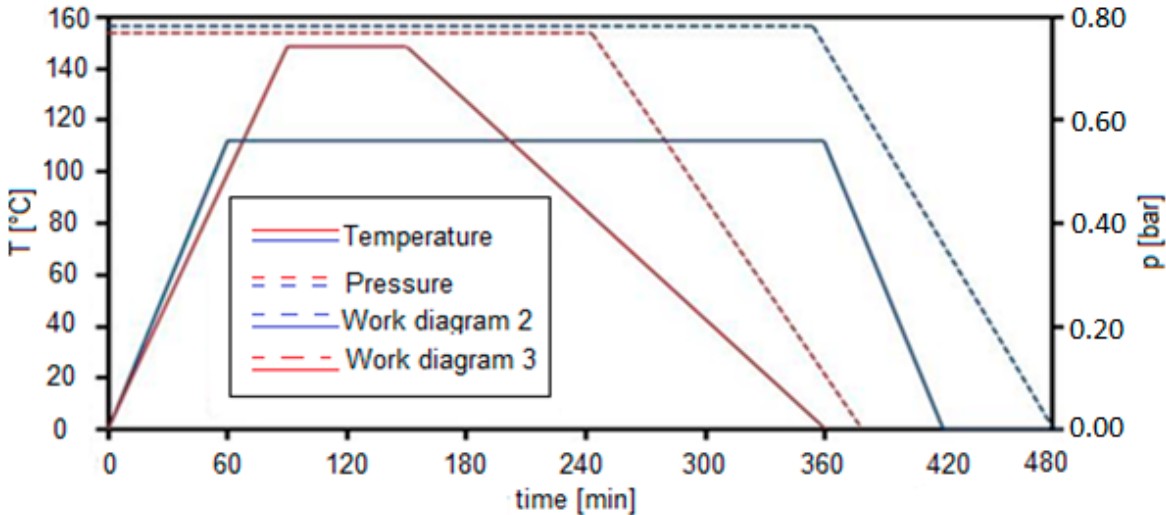

**Figure 17.** Alternative curing cycles.

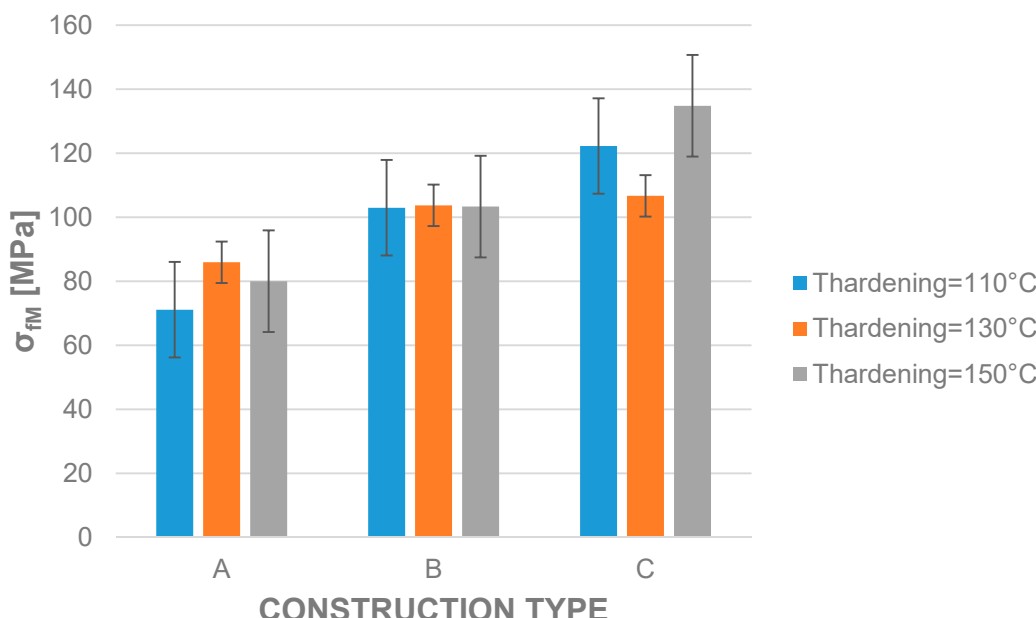

**Figure 18.** Influence of curing cycles on the bending strength of the A, B, and C constructions (used core: honeycomb C2-3.2-80/L orientation).

In terms of the influence on bending modulus, Curing Cycle 2 seems to be most suitable. The A and B constructions produced in this mode achieved the highest values of bending moduli as regards the given construction group, while at the C construction, the achieved bending modulus was almost identical to Curing Cycle 3. On average, the values of bending moduli produced within this mode (110 °C) were higher by 5.51% as compared to Curing Cycle 1.

A summary of the three-point bending measurements is given in Table 3.

**Table 3.** Three-point bending results.

| Core Type C2-3,2-64 $t_c$ = 8 mm | Unit | Construction A | | Construction B | | Construction C | |
|---|---|---|---|---|---|---|---|
| Direction | | L | W | L | W | L | W |
| $E_{ef}$ | (MPa) | 5661 | 5148.25 | 7191 | 6636.5 | 7859 | 7245.5 |
| $\sigma_{fM}$ | (MPa) | 67.47 | 59.9 | 90.7 | 69.52 | 101.25 | 67.02 |
| $F_{max}$ | (N) | 1001.25 | 889.1 | 1487.25 | 1139.75 | 1849 | 1225.25 |
| W at $F_{max}$ | (J) | 3301.99 | 2888.6 | 4997.98 | 3160.87 | 5847.21 | 2771.01 |

## 4. Impact Testing

For these reasons, it is necessary to define the influence of core material types and thicknesses as well as the variability in the thicknesses of coatings for the toughness of such materials exposed to impact stressing. In the evaluations of the impairment scope and mechanism of the samples exposed to falling bodies, it is possible to define impairment hypotheses and, at the same time, to adopt follow-up measures for the construction of sandwich materials.

The set of samples was tested according to the ISO 6603-2 Standard, using the ZWICK 1456 test equipment (ZwickRoell, Brno, Czech Republic). A hemisphere-shaped impactor (10 mm diameter) was selected, and its falling rate ranged between 1.55 to 1.65 m·s$^{-1}$; the test temperature was kept at $20 \pm 2$ °C. In the test arrangement, the maximum impact force was evaluated. Two levels of impact energy were chosen: 30 J and 60 J.

In the course of impact tests, the sandwich constructions with variable coating thicknesses (A, B, and C constructions) were evaluated with a primary goal of determining the influence of coating thickness and core type on the impact resistance of sandwich constructions.

In the sandwich constructions with thicknesses of ~10 mm, the influence of six core types in total—three types of Nomex honeycombs of density 32 kg·m$^{-3}$, 48 kg·m$^{-3}$, and 64 kg·m$^{-3}$; one type of aluminium honeycomb of 77 kg·m$^{-3}$ density; and one type of AIREX T90-100 polymer foam—was studied. The maximum values of the force developed in the impact course are shown in Figure 19.

In terms of the thickness influence, an increase in the impact force can be noted with increasing thickness. This effect can be observed, at both impact energy levels, in all constructions produced of Nomex cores and polymer foams. A perceptible increase in the impact force can be observed at a higher level between the A and B constructions rather than between the B and C constructions. In addition, significant differences were noted between the densities and/or mesh sizes of the honeycombs used.

As regards the core material types used, the highest impact force values are shown in the constructions with polymer cores as compared to all other core materials; on the other hand, for the constructions with aluminium honeycombs, the lowest values were recorded. As regards this core type, the influence of coatings could not be reasonably evaluated.

It can be surmised from the measured results that the value of maximum impact force is significantly influenced by the used core types. Consequently, the resistance can be further increased by changing the coating structure, which is mostly efficient in constructions containing polymer foams (i.e., the increase in resistance of the A type construction compared with the B type construction by 36.86% at the impact force of 30 J, or by almost 84% at the impact force of 60 J), which is clearly evident even from the graphic recording of the test progress.

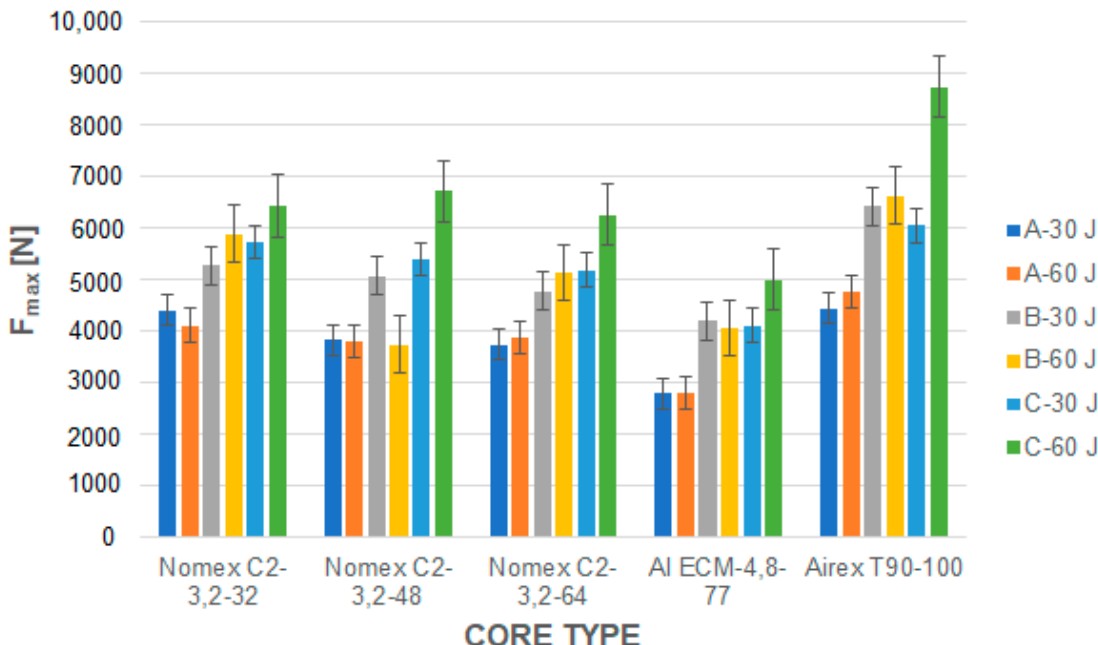

**Figure 19.** An overview of maximum forces in the impact tests. Sandwich constructions with cores of $t_c = 9$ mm and $t_c = 10$ mm thicknesses.

After the test, sandwich structures were visually inspected for defects. These defects usually result in impact tests and they include defects on upper and lower coatings/apart from the C type structures, core deformations, defects on the coating–core interface, and coating delamination (peeling).

Defective samples with Nomex honeycombs and polymer foams show narrowly localized defects of circular cross sections approximately copying the impacting body. In the case of aluminium honeycomb structures, the scope of damage is larger due to higher damaging of the coating–core interface. The composite coating itself shows damaged points in the form of broken fibers at two mutually perpendicular axes, which corresponds with prepreg placements in the coating with 0/90° orientation. Furthermore, delamination of individual layers is visible in the damaged area.

In the following phase of penetration tests, composite sandwich materials with 20 mm thick cores were tested. The test included the following 20 mm thick core materials; C2-4.8-32 Nomex honeycomb, AIREX T90-100 polymer foam, and ECM 4.8-77 aluminium honeycomb. The results and comparisons for sandwich constructions with cores up to 10 mm thickness are shown in Figure 20.

The highest values of impact forces were obtained with the constructions with Nomex honeycomb cores and, on the contrary, the lowest values were found with the constructions with aluminium honeycomb cores, and this is the case for cores up to 10 mm and even 20 mm. The significant drop in maximum stresses of the constructions with polymer cores was noted. For 20 mm thick cores, the coating thickness influence was always positive.

The scope and type of damage in the constructions with 20 mm thick cores are almost identical as in the previous case. For the aluminium honeycombs, a substantial separation of the lower coating is visible. At 13 mm thick Nomex honeycomb cores (C2-3.2-64), the construction defects are characterized with a pronounced shear disturbance approximately in the core center.

In detail, it is possible to see the core thickness influence in Figure 21, where the plotted maximum puncture force for constructions with 8, 10, 13, and 20 mm thicknesses is shown. In the framework of these thicknesses, all three construction types (A–C) were tested at two impact energies: 30 J and 60 J.

There was an apparent drop in the impact resistance in the constructions with 13 mm thick cores (Figure 21), while in the constructions with 20 mm thick cores, an increase could be found again, where these (20 mm thick) constructions show the maximum values within the whole frame of observation. The drop in the resistance with increasing thickness of constructions (with 13 mm

thickness) can be explained by relatively significant reinforcements of cell walls of thinned cores with resin from the phenolic coating. On the other hand, with 20 mm thick cores, the honeycomb deformation property takes place probably to a greater degree. In addition, it is also possible to observe the fact that the honeycombs of lower densities (10 mm and 20 mm thicknesses) show better parameters than the honeycombs of 64 kg·m$^{-3}$ density and smaller mesh site (8 and 13 mm thicknesses).

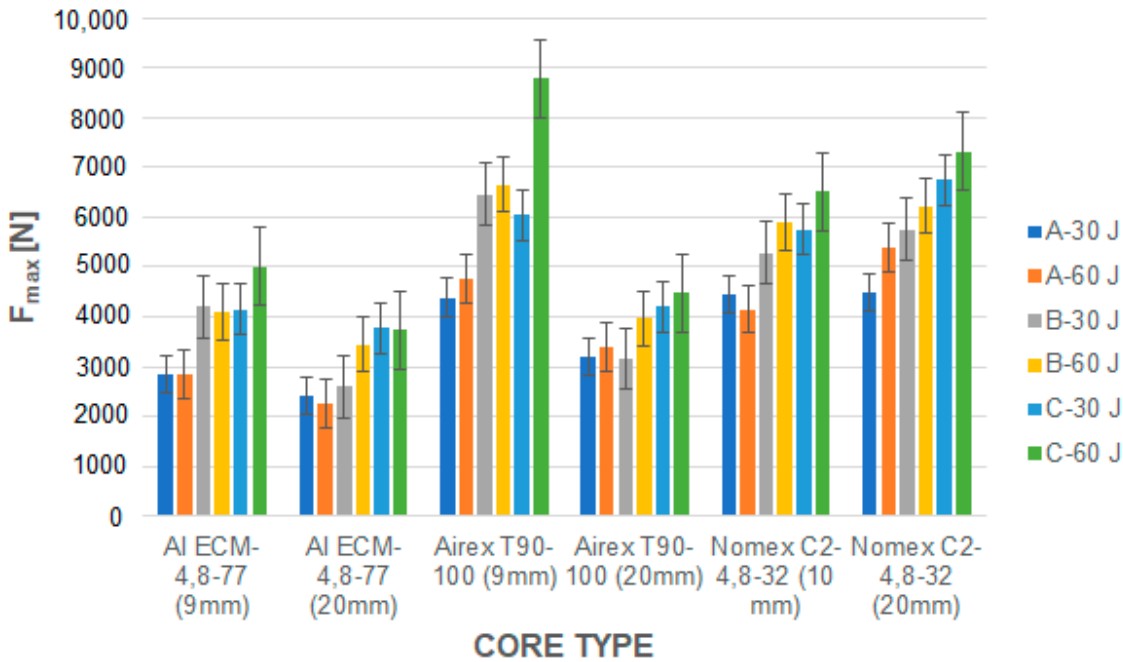

**Figure 20.** A survey of maximum forces at the impact test. Sandwich constructions with $t_c$ = 20 mm thick cores. A comparison with the structures $t_c$ = 10 mm thick cores.

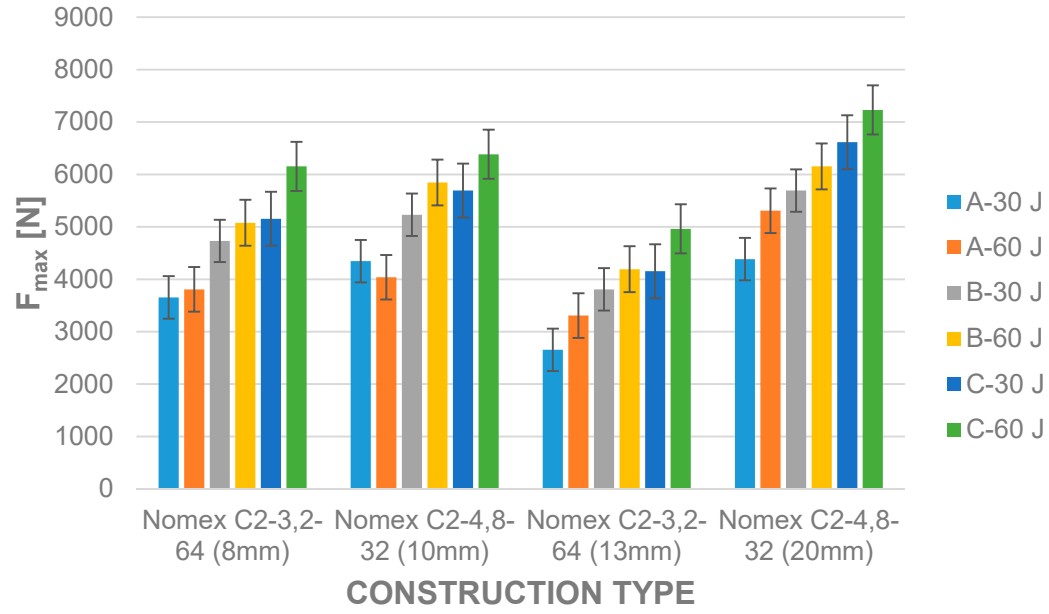

**Figure 21.** Influence of core thicknesses; the honeycomb used of various geometry C2-3.2-64 (W orientation).

In all groups with identical cores of the given thickness, a positive influence of increasing the coating thickness for the impact resistance can be observed as it can be seen from Tables 4–6.

**Table 4.** Results of impact testing till $t_c < 10$ mm.

| | Construction 1 HC Nomex $t_c$ = 8 mm C2-3.2-64 | | | Construction 2 AL Alloy $t_c$ = 9 mm 3.2-77 | | | Construction 3 PET Foam $t_c$ = 9 mm AIREX T90-100 | |
|---|---|---|---|---|---|---|---|---|
| **Sheet Construction** | A | B | C | A | B | C | A | B |
| $F_{max}$ **(N)** | 3674.8 | 4755 | 5171.9 | 2778.5 | 4170.8 | 4096.6 | 4390.8 | 6427.3 |
| $F_p$ **(N)** | 1837.4 | 2377.5 | 2586 | 1389.2 | 2085.4 | 2048.3 | 2195 | 3213 |
| $E_p$ **(J)** | 18.8 | 31.05 | 32.05 | 21.05 | 33.3 | 32.04 | 22.42 | 32.12 |

**Table 5.** Results of impact testing with different types of core material Nomex.

| | Construction 1 HC Nomex $t_c$ = 8 mm C2-3.2-64 | | | Construction 2 HC Nomex $t_c$ = 9 mm C2-3.2-48 | | Construction 3 HC Nomex $t_c$ = 10 mm C2-6.4-32 | | Construction 4 HC Nomex $t_c$ = 13 mm C2-3.2-64 | | |
|---|---|---|---|---|---|---|---|---|---|---|
| **Sheet Construction** | A | B | C | A | B | A | B | A | B | C |
| $F_{max}$ **(N)** | 3674.8 | 4755 | 5171.9 | 3815.9 | 5031.2 | 4393.6 | 5266.9 | 2335.2 | 3005.4 | 3410.7 |
| $F_p$ **(N)** | 1837.4 | 2377.5 | 2586 | 1907.95 | 2515.6 | 2196 | 2633.4 | 1167.59 | 1502.71 | 1705.38 |
| $E_p$ **(J)** | 18.8 | 31.05 | 32.05 | 19.59 | 31.7 | 14.15 | 24.01 | 23.73 | 33.55 | 35.90 |

**Table 6.** Results of impact test with different Nomex core.

| | Construction 1 HC Nomex $t_c$ = 8 mm C2-3.2-64 | | | Construction 2 HC Nomex $t_c$ = 9 mm C2-3.2-48 | | Construction 3 HC Nomex $t_c$ = 10 mm C2-6.4-32 | | Construction 4 HC Nomex $t_c$ = 13 mm C2-3.2-64 | | |
|---|---|---|---|---|---|---|---|---|---|---|
| **Sheet Construction** | A | B | C | A | B | A | B | A | B | C |
| $F_{max}$ **(N)** | 3674.8 | 4755 | 5171.9 | 3815.9 | 5031.2 | 4393.6 | 5266.9 | 2335.2 | 3005.4 | 3410.7 |
| $F_p$ **(N)** | 1837.4 | 2377.5 | 2586 | 1907.95 | 2515.6 | 2196 | 2633.4 | 1167.59 | 1502.71 | 1705.38 |
| $E_p$ **(J)** | 18.8 | 31.05 | 32.05 | 19.59 | 31.7 | 14.15 | 24.01 | 23.73 | 33.55 | 35.90 |

## 5. Conclusions

The presented case study is focused on useful experimental findings about the mechanical properties of sandwiches that are influenced by materials as well as layer arrangements in the structures of such composites. Experimentally, it was found that the specific strength is positively influenced only by the construction core. In all three presented hybrid constructions, it is possible to see significant drops in bending strengths as well as in effective bending moduli. If the strength characteristics of sandwich constructions are considered in terms of honeycomb densities, then their significant positive influence on the bending strength will be evident. On the other hand, the use of higher density honeycombs brings about, with the density increase, a much less effect to the bending modulus. The highest differences between the bending strength in the L and W directions were recorded for the honeycomb density of 64 kg·m$^{-3}$. The effective bending modulus was higher for all constructions in the L direction as compared to the W direction. The honeycomb orientation influence on bending characteristics was evaluated even in terms of another parameter of honeycomb geometry, and this was the mesh size. In this case, the evaluated honeycombs were of constant density of 64 kg·m$^{-3}$ with mesh sizes of 3.2, 4.8, and 6.4 mm. Similarly, as in the previous case, the bending strength in the L direction was always higher than that in the W direction.

Regarding the influence of thicknesses, it is possible to observe an increase with increasing thicknesses of the coating. As concerns the core material type used in the composite, the highest impact strength occurs in the constructions with a polymer core as compared to all other core materials, while, on the contrary, in the constructions with aluminium honeycombs, the obtained values were at their lowest. The highest values of impact forces are found in the constructions with Nomex honeycomb cores, while, on the other hand, the constructions with aluminium honeycomb returned the lowest values, namely, in the case of 10 mm and 20 mm thick cores. A significant drop in the maximum stress in constructions with polymer cores is remarkable. Even in the case of 20 mm thick cores, the influence of coating thickness is always positive. Within the framework of all groups of the same core of the

given thickness, a positive influence on the impact resistance can be observed with increasing the coating thickness.

The sandwich material properties were also influenced by thermal modes used in their processing. In terms of influencing the bending modulus, Curing Cycle 2 seems to be the most favorable.

**Author Contributions:** Conceptualization, P.K.; methodology, P.K. and Z.K.J.; software, Z.K.J.; validation, M.G. and I.R.; formal analysis, Z.K.J.; investigation, I.R.; resources, P.K.; data curation, P.K. and Z.K.J.; writing—original draft preparation, P.K.; writing—review and editing, P.K. and I.R.; visualization, I.R.; supervision, P.K.; project administration, M.G.; funding acquisition, M.G. All authors have read and agreed to the published version of the manuscript.

**Funding:** This research received no external funding.

**Acknowledgments:** This work was supported by Ministry of Education, Youth and Sports, Czech Republic in the framework of the projects SP2020/18, SP2020/61, and SP2020/39, and by the Ministry of Education, Science, Research and Sport of the Slovak Republic in the framework of the project VEGA 1/0717/19.

**Conflicts of Interest:** The authors declare no conflict of interest. The funders had no role in the design of the study; in the collection, analyses, or interpretation of data; in the writing of the manuscript; or in the decision to publish the results.

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
