# Peer review of "Case Study of Chosen Sandwich-Structured Composite Materials for Means of Transport"

_coatings, doi:10.3390/coatings10080750_

Round 1

Reviewer 1 Report

This article shows the mechanical properties modulus of elasticity and maximum force at impact, obtained experimentally for different configurations of honeycomb sandwiches. Different core and coating compositions and different geometries are analyzed.
The work is very extensive and contains a large number of experimental results, but needs mayor revision to be published in Journal Coatings.

Comments
- The title is ambiguous, it is not known what is analyzed and what type of sandwich is studied
- In the abstract, the content of the work and its contribution to knowledge and novelty must be clear. Also, the work should be limited to the types of sandwiches studied, thick with a honeycomb core
- The authors must explain why they carry out static and impact analysis when this type of sandwich structure is effective in vibroacoustic attenuation and therefore dynamic testing would be necessary
- The paper should contain clear and concise objectives, and then arrange the results to meet the objectives. Analysis of the influence of coating thickness and coating orientation is considered to be of interest
- All the work needs to be ordered and it is even suggested that the most relevant results be selected, because the large number of graphs makes them difficult to understand.
- The notation must be unique, equivalent tables to Table 1 (coatings) and Table 2 (core) must be included, with the configurations that are analyzed in the paper with a unique notation throughout the paper.
- References should be reviewed, and include more recent contributions. There are no references after 2015
- All graphics should have a similar format, and it should be assessed whether they provide knowledge
- The processing conditions of each of the analysed certifications must be included and explained in the ASTM C-393 standard
- The experimental tests, equipment, procedure and treatment of results and the standards used should be described in detail.
- The number of specimens of each type of sample tested and the dimensions should be described
- The number of samples tested and the experimental errors should be described.

Reviewer 2 Report

The papers concerns some experimental analysis of specific honeycomb structures. In this aspect, it lacks experimental error analysis and statistical processing of the collected results. A lack of additional experimental statistics makes further engineering reliability analysis completely impossible. 

The manuscript needs some serious editing works, like references section as well as some figures with skew text below horizontal axes. 

An overview of the literature is rather skimpy and should be extended with some theorerical and numerical approaches to multiscale analyses (cf.M. Kamiński, Multiscale homogenization of n-component composites with semi-elliptical random interface defects. Int. J. Sol. & Struct. 42 (11-12), 3571-3590, 2005). 

The sentences included in conclusions (cit.) : "Today, the obtained knowledge is already utilisedutilized in the industrial practice: in construction designs, adjustment of process parameters, in manufacturing procedures and applications of concrete sandwich products in the industry of transport means. These products, featured by their low weight, contribute to a better fuel economy as well as improved cost efficiency of railway vehicles, and through this to a reduced environmental burden created by the mass transport." are trivial and do not follow the analysis contained in this work. 

Reviewer 3 Report

There are new ways to measure fracture toughness of thin films and coatings which should be cited:

1- A machine learning approach to fracture mechanics problems, X Liu, CE Athanasiou, NP Padture, BW Sheldon, H Gao, Acta Materialia 190, 105-112

2- A new method to evaluate the fracture toughness of thin films, Z Xia, WA Curtin, BW Sheldon, Acta materialia 52 (12), 3507-3517

Overall the paper is good. I suggest it for publication.

Round 2

Reviewer 1 Report

This paper shows the modulus of elasticity in static response and maximum forcé to the impact, obtained experimentally for different configurations of honeycomb sandwiches. Different core, skins, geometries are analyzed.

The work is very extensive and contains a large number of experimental results. With the changes made by the authors the content has improved, but still needs an in-depth review to be published in Journal Coatings.

In particular, the authors must take into account those important points that have not been modified:

- The title is ambiguous, it is not known what is analyzed and what type of sandwich is analyzed

- The abstract must be rewritten. The content of the work and its contribution to knowledge and novelty must be clear. If 3 point bending (ASTM C-393) and impact (ISO 6603-2 Standard) tests are performed, both standards should be mentioned. Also, the work should be limited to the types of sandwiches studied, thick with a honeycomb core

- The paper should contain clear and concise objectives.

- It should describe in detail the experimental tests, equipment, procedure and treatment of results and the standards that have been used, for static (ASTM C-393) and impact tests (ISO 6603-2 Standard).

Author Response

This paper shows the modulus of elasticity in static response and maximum forcé to the impact, obtained experimentally for different configurations of honeycomb sandwiches. Different core, skins, geometries are analyzed.

The work is very extensive and contains a large number of experimental results. With the changes made by the authors the content has improved, but still needs an in-depth review to be published in Journal Coatings.

In particular, the authors must take into account those important points that have not been modified:

  1. The title is ambiguous, it is not known what is analyzed and what type of sandwich is analyzed

We have changed the title of article as follows:

Case Study of Chosen Sandwich – Structured Composite Materials for Means of Transport

  1. The abstract must be rewritten. The content of the work and its contribution to knowledge and novelty must be clear. If 3 point bending (ASTM C-393) and impact (ISO 6603-2 Standard) tests are performed, both standards should be mentioned. Also, the work should be limited to the types of sandwiches studied, thick with a honeycomb core

We have changed abstract as follows:

Modern means of transport increasingly utilize sandwich constructions. Among other things, the reasons for such state of affairs include the reduced weight of means of transport, and through this, better fuel economy as well as price. This work is dedicated to a systematic experimental study of the influence of various materials and sandwich designs on their mechanical properties. In the framework of experiments, sandwich-structured composites were exposed to two types of stressing: static as well as impact stressing. The testing of prepared samples was performed according to ASTM C-393 Standard, dealing specifically with the bending behavior of sandwich composite constructions and impact testing under the scope of ISO 6603-2 Standard test. In this article we deal with static and impact testing of the eight types of core materials, two types of coatings, two types of surface finishes, and two types of resins with a special emphasis on their use in constructions of some exterior or interior components of transport means.

  1. The paper should contain clear and concise objectives.

We have changed the objectives:

The objectives of the paper was to find whether the sandwich-structured coated composites can be used as parts in constructions of some exterior or interior components of transport means. Therefore chosen sandwich-structured coated composites were mechanically tested and bending strength, specific bending strength (ratio of bending strength to density), bending modulus and specific bending modulus (ratio of bending modulus to density) at static testing under ASTM C-393 framework and impact testing at 30J and 60J impact energy with computed mean values of maximal obtained force. Computation of specific bending strength and specific bending modulus is crucial for the application of materials in the means of transport because of lowering price and better fuel economy

  1. It should describe in detail the experimental tests, equipment, procedure and treatment of results and the standards that have been used, for static (ASTM C-393) and impact tests (ISO 6603-2 Standard).

We changed:

Eight types of core materials, two types of coatings, two types of surface finishes, and two types of resins were experimentally tested by 3pt bending test according to ASTM C-393 standard and impact tested according to ISO 6603-2 Standard test. The tests were provided by five samples of the same construction and of the same materials and bending strength σFM, bending modulus E were computed from the stress-strain diagram. From 5 values of this mechanical properties we have computed the mean value and standard deviation (which are plotted in Figures) using the ZWICK 1456 test equipment (ZwickRoell, Brno, Czech republic). In the second step the materials were impact tested according to ISO 6603-2 norm and maximal force Fmax obtained at 30J and 60J impact energy were measured. The 5 values for each material were subsequently processed by descriptive statistics parameters namely mean value and standard deviation which are also plotted in the Figures 18, 19, 20. From these two statistical parameters, we have computed the coefficient of variation, which for all measuring samples and all measured properties was lower than 15%, which is an acceptable level of error. The figures show statistically significant differences; therefore, ANOVA tests results are not shown in this study. The shape of the samples was adapted to the requirements of the relevant standard.

Reviewer 2 Report

The Authors have modified their manuscript accordingly and it is now ready for publication in this journal.